# Obtainment and Characterization of Hydrophilic Polysulfone Membranes by N-Vinylimidazole Grafting Induced by Gamma Irradiation

**DOI:** 10.3390/polym12061284

**Published:** 2020-06-04

**Authors:** Elizabeth Vázquez, Claudia Muro, Javier Illescas, Guillermina Burillo, Omar Hernández, Ernesto Rivera

**Affiliations:** 1Tecnológico Nacional de México/Instituto Tecnológico de Toluca, Avenida Tecnológico S/N Col. Agrícola Bellavista, C.P. 52140 Metepec, Mexico; evazquezr@toluca.tecnm.mx (E.V.); fillescasm@toluca.tecnm.mx (J.I.); ohernandeza@toluca.tecnm.mx (O.H.); 2Laboratorio de Química de Radiaciones en Macromoléculas, Instituto de Ciencias Nucleares, Universidad Nacional Autónoma de México, Cd. Universitaria, C.P. 04510 CDMX, Mexico; burillo@nucleares.unam.mx; 3Instituto de Investigaciones en Materiales, Universidad Nacional Autónoma de México, Avenida Universidad #3000 Colonia UNAM, Delegación Coyoacán, C.P. 04510 CDMX, Mexico; riverage@unam.mx

**Keywords:** membrane grafting, polysulfone, N-vinylimidazole, gamma irradiation, hydrophilic surface, desalination

## Abstract

Polysulfone (PSU) film and N-vinylimidazole (VIM) were used to obtain grafted membranes with high hydrophilic capacity. The grafting process was performed by gamma irradiation under two experiments: (1) different irradiation doses (100–400 kGy) and VIM 50% solution; (2) different concentration of grafted VIM (30–70%) and 300 kGy of irradiation dose. Characteristics of the grafted membranes were determined by Fourier transform infrared spectroscopy (FTIR), scanning electron microscopy (SEM), contact angle, swelling degree, desalination test, thermogravimetric analysis (TGA) and differential scanning calorimetry (DSC). Both experiments indicated that the absorbed dose 300 kGy and the VIM concentration, at 50% *v/v*, were effective to obtain PSU grafted membranes with 14.3% of grafting yield. Nevertheless, experimental conditions, 400 kGy, VIM 50% and 300 kGy, VIM 60–70% promoted possible membrane degradation and VIM homopolymerization on the membrane surface, which was observed by SEM images; meanwhile, 100–200 kGy and VIM 30–50% produced minimal grafting (2 ± 0.5%). Hydrophilic surface of the grafted PSU membranes by 300 kGy and VIM 50% *v/v* were corroborated by the water contact angle, swelling degree and desalination test, showing a decrease from 90.7° ± 0.3 (PSU film) to 64.3° ± 0.5; an increment of swelling degree of 25 ± 1%, and a rejection-permeation capacity of 75 ± 2%. In addition, the thermal behavior of grafted PSU membranes registered an increment in the degradation of 20%, due to the presence of VIM. However, the normal temperature of the membrane operation did not affect this result; meanwhile, the glass transition temperature (T_g_) of the grafted PSU membrane was found at 185.4 ± 0.5 °C, which indicated an increment of 15 ± 1%.

## 1. Introduction

Membrane filtration technology is widely used for various industrial separation, concentration, and purification applications, because they are effective processes that require less energy and steps than typical separation operations, such as distillation, extraction and drying. Nonetheless, the major limitation to the effective application of membrane technology is fouling, because this phenomenon limits its rejection-permeation capability [1].

Fouling appearance is inevitable in membrane processes, as it decreases water productivity, deteriorates permeate quality, and shortens membrane lifespan [2,3,4,5]. Fouling level and types depend on membrane operational factors, hydrodynamic conditions and membrane characteristics [6]. 

From these mentioned aspects, several alternatives have been studied to decrease the fouling phenomenon. Mainly, the design and preparation of membranes with hydrophilic properties are currently the subject of research in several reports.

Methods to obtain hydrophilic membranes include the graft of polymers with antifouling characteristics. These materials are covalent bonded to the main chain of a polymeric matrix [7,8,9,10]. The graft of antifouling polymers enhances membrane permeability, performance and membrane lifetime, avoiding undesired interactions between both the membrane material and the feed solution [11,12]; meanwhile, the operation costs and environmental impacts in the membrane applications are also reduced by this process [12,13,14].

The most common materials used as matrices for grafting processes are polysulfone (PSU), polyamide (PA) and polypropylene (PP). In turn, antifouling polymers, such as betaines [15], poly(2-hydroxyethyl methacrylate) (PHEMA) [16], polyacrilic acid (PAAc), 1-aminopyrene (AP) [17], 2,4,6-tri(dimethylaminomethyl)-phenol (TDAP), [18], hydroxyethyl acrylate(HEA) [19], and vinylimidazole (VIM) [20] have been tested as grafting agents, because they can improve the hydrophilicity and porosity of the membranes [21,22,23]. In addition, polymers such as PVIM are also used as antimicrobial agents, because they avoid membrane biofouling, enhancing membrane rejection-permeability and membrane efficiency [24,25,26].

The grafting process is often realized through transfer agents (RAFT) by conventional chemical methods; however, the use of irradiation sources are of great interest, because they promote graft polymerization, due to the fast formation of active sites on the substrate surface and in the material matrix during the grafting process [27,28,29,30,31,32].

Mostly, membrane grafting studies have been focused on the area of polymer electrolyte membrane fuel cells (PEMFCs) [33] (Vijayakumar and Nam, 2018) and the modification of exchange membranes. Information related to this field has been reported in Masuelli et al. [34]; authors described the synthesis of cationic poly(vinylidene fluoride) (PVDF) membranes by glycidyl methacrylate (GMA) and ethylene glycol dimethacrylate (EDMA). Additionally, Guo et al. [35] used poly(vinylbenzyl chloride) (PVBC) and VIM as graft agents into a poly(vinyl alcohol) PVA matrix; whereas, Wu et al. [36] obtained imidazolium-type hybrid alkaline anion exchange membranes based on a poly(2-bromomethyl-6-methyl-1,4-phenylene oxide) matrix, employing UV irradiation and a mixture of VIM with hollow mesoporous silica spheres.

UV graft polymerization has also been used to increase the hydrophilicity of commercial membranes for their applications in bioseparations and biomedical applications [37]. Nevertheless, publications on grafted membranes for desalination and food industry are scarce; particularly, the reports on PSU membranes manufacturing by induced irradiation were found by Cheng et al. [38], where authors used UV light as an exposure source for the grafting development; whereas, Davari et al. [39] also studied the grafting of VIM and betaine groups onto a polyamide membrane by gamma irradiation. As a result, grafting agents enhance the antifouling ability to resist non-specific protein adsorption at neutral and alkaline pH, including the inhibition of bacterial progress.

For these reasons, in this work, the induction of gamma irradiation dose and monomer concentration of VIM were studied to obtain PSU grafted membranes with high hydrophilic capacity. The characterization techniques, such as Fourier-transform infrared spectroscopy (FTIR), scanning electron microscopy (SEM), contact angle, swelling degree properties, desalination performance tests, thermogravimetric analysis (TGA) and differential scanning calorimetry (DSC) were used to determine the most important characteristics of the grafted PSU membranes. Therefore, the results of this study will contribute to apply gamma radiation as a technique of membrane grafting to improve its hydrophilic characteristics.

## 2. Materials and Methods

### 2.1. Materials

N-vinylimidazole (VIM) reagent (99%) was purchased from Sigma-Aldrich Co. (Toluca, Mexico), methanol and acetone were purchased from JT Baker Avantor Performance Materials S.A. de C.V. (Ecatepec, Mexico), and polysulphone (PSU) film Udel™/Ultrason™ (0.1 mm thickness, 1.24 g·cm^−3^) was supplied by Goodfellow Cambridge Limited (Huntingdon, England).

### 2.2. Grafted Membranes Preparation

Grafted membranes were obtained by VIM as the grafting agent and PSU films as the polymer matrix. The grafting process was carried out by direct irradiation in two different experiments. (1) PSU films were put in contact with VIM, in a solution concentration of 50% (*v/v*); afterwards, they were exposed to gamma irradiation at different radiation doses between the range of 100–400 kGy. (2) PSU films were exposed to a constant radiation dose, according to the obtained results in the first experiment, and with a variation of the VIM concentration, employing solutions of 30, 50, 60, and 70% (*v/v*). The radiation doses range, from 100 to 400 KGy, was selected because doses below 100 kGy did not promote the grafting polymerization of VIM; in addition, the radiation resistance of PSU is up to 400 kGy.

For both experiments, pristine PSU films (4 × 1 cm) were previously washed with acetone and dried until a constant weight was achieved; afterwards, films were placed into glass ampoules containing 8 mL of VIM monomer solution. Ampoules were degassed by bubbling argon for 20 min; then, ampoules were sealed and exposed to a gamma irradiation source of ^60^Co, in a Gammabeam 651-PT irradiator at the Universidad Nacional Autónoma de México (UNAM), with a dose rate of 9.71 kGy/h. After irradiation, the grafted membranes were washed with deionized water for 72 h and methanol for 24 h, to remove the residual monomer (VIM) and the formed homopolymer (PVIM) during the grafting process. Finally, membranes were dried in a vacuum oven at 60 °C until a constant weight was obtained.

### 2.3. Characterization of PSU Grafted Membranes

Grafted membranes were characterized according to their superficial morphology, their chemical structure, their swelling degree by means of water contact angle, their thermal decomposition and their thermodynamic transition properties.

The grafting percentage of PVIM onto PSU membranes was calculated from the weight of the pristine membrane (W_0_) and the grafted membrane (W_g_), as indicated in Equation (1).
(1)$$\mathrm{Grafting}\;\left(\%\right)=\frac{100\;\left(W_{g}-{\;W}_{0}\right)}{W_{0}}$$

The chemical structure of the grafted membranes was determined by means of FTIR-ATR spectroscopy, using a VARIAN IR-640 fitted with an ATR universal accessory (Agilent Technologies, CA, USA), with a resolution of 16 scans. Bands of the analyzed spectra, at a wavenumber range of 4000–600 cm^−1^, were determined.

The surface morphology of the grafted membranes was evaluated by means of scanning electron microscopy (SEM) using a JSM-7600F (JEOL Ltd., Tokyo, Japan) instrument. Samples were sputter coated with gold for 45 s; then, they were evaluated with a secondary emission detector at a voltage of 0.50 kV, at 25,000× and a working distance (WD) of 5.5 mm.

Water contact angles from the grafted membranes were measured from distilled water drops onto dry films at room temperature by means of a Kruss DSA 100 drop shape analyzer (Matthews, NC, USA).

The swelling degree from the grafted membranes was determined by soaking some pieces of them in distilled water at a constant temperature, weighing each membrane at several different times until their constant weight was achieved. Meanwhile, the swelling degree data of the grafted membranes were determined by the immersion of membranes in assorted buffers with different pH values, from 2 to 12. The swelling degree was calculated gravimetrically, applying Equation (2), where W_s_ and W_d_ represent the weights of the swelled and dried grafted membranes, respectively.
(2)$$\mathrm{Swelling}\;\mathrm{degree}\;\left(\%\right)=\frac{100\;[\left(W_{s}-{\;W}_{d}]\right)}{W_{d}}$$

Thermal decomposition monitoring was performed at a heating rate of 10 °C·min^−1^ from 25 to 800 °C under a controlled nitrogen atmosphere by a TGA Q50 analyzer (TA Instruments, New Castle, DE, USA).

Thermodynamic transition properties from grafted membranes were obtained by differential scanning calorimetry (DSC) analysis, at a heating rate of 10 °C·min^−1^ from −80 to 250 °C. Samples were analyzed under nitrogen atmosphere using a DSC 2010 calorimeter (TA Instruments, New Castle, DE, USA), employing a temperature range between 25 to 250 °C.

Membranes desalination test was based in the membrane rejection-permeation of one solution containing 1000 mg/L of sodium chloride (NaCl) at 25 °C, pH 6 and 15 bar of pressure. The rejection-permeation of NaCl and water was measured by flow rejection and permeation and total solids from each stream for the grafted membrane (previously selected with the highest hydrophilicity). Fluxes were calculated from the volume of rejection-permeation source per unit of time and area by their collection in calibrated vessels at regular time spans.

The desalination test was carried out on a dead-end filtration system of flat membrane module, using flat grafted membranes with an average area of 19.63 cm^2^. The grafted membrane was immersed in deionized water overnight before the flux measurement was undertaken.

## 3. Results and Discussion

### 3.1. Grafting Yield of VIM onto PSU Membranes

Experiments 1 and 2 displayed data about the effect of gamma irradiation doses (kGy) and VIM concentration agent to obtain the grafted PSU membranes (PSU-VIM). Figure 1a,b describe the grafting yield percentage (%) of VIM onto PSU membranes by these experiments. Herein, it was assumed that gamma irradiation conditions promoted the formation of free radicals and the production of the active sites of reaction in the PSU film; meanwhile, the VIM polymerization onto the PSU membranes surface was allowed by the chosen irradiation doses to obtain grafted membranes with different yields.

The mechanism of the grafting process was attributed to the break of the double bonds of the vinyl monomer of the VIM, which reacts with the surface radicals from the PSU membrane [31]. Therefore, the grafted membranes were predictable in the range of the reverse osmosis process (RO), due to their origin (PSU film). This assumption was confirmed by the desalination test, of which the results are presented below.

Figure 1a (from experiment 1) showed that the grafting yield of VIM increased according to the irradiation doses, achieving 2 ± 0.5% (100 kGy) to a maximum of 14.5 ± 1% (400 kGy). Herein, it is notorious that the yield percentage of this last condition was close to 300 kGy (13.8 ± 3%), indicating that irradiation doses >300 kGy did not produce high yields.

Figure 1b displayed the effect of VlM concentration solution (30%, 50%, 60% and 70 % *v/v*) with the gamma irradiation dose of 300 kGy on the grafting yield, showing an exponential tendency between the VIM grafting yield and the VIM concentration. Specifically, the grafting of VIM 30% (*v/v*) did not demonstrate any significant increase (3.1 ± 0.5%); whereas, higher concentrations of VIM (50–70% *v/v*) produced the highest yield of 13.5 ± 5%, 29 ± 3%, and 42 ± 2%, respectively.

From the above results, Experiment 2 indicated that both the irradiation dose of 300 kGy and the VIM concentration (50–60% *v/v*), were adequate conditions to obtain high grafting yields of PSU-VIM membranes. This result was assumed because grafting with doses under 300 kGy produced less grafting, and with a dose of 400 kGy, a VIM homopolymer layer is most probably produced. This same phenomenon was observed in the grafting of membranes with the highest VIM concentration, 70% (*v/v*), that will be discussed later.

Comparable tests were found in reports on grafting membranes; however, values of the grafting yields of VIM onto PSU membranes were different. The variances were attributed to the graft agent, the grafting matrix, the irradiation source, and the irradiation doses. Melendez-Ortiz et al. [28] grafted VIM onto silicone rubber by gamma irradiation to obtain antimicrobial properties. An increase of the grafting yield with respect to the concentration and the irradiation gamma dose was observed, achieving values up to 200%. Costoya et al. [40] grafted glycidyl methacrylate (GMA) onto poly(vinyl chloride) (PVC). A pre-irradiation dose of 40 kGy produced a yield of 120%; whereas, Caner et al. [41] chemically grafted poly(N-vinylimidazole) onto chitosan via ceric ion initiation, achieving 140% of grafting yield. In the same context, 2-Hydroxyethyl methacrylate (HEMA) and N-vinylcaprolactam (NVCL) were grafted onto PP films by the pre-irradiation oxidative method (10 and 70 kGy), showing 80% of yield [42].

### 3.2. Chemical Structure of the Grafted PSU Membranes

FTIR-ATR analysis provided the chemical structure of the grafted PSU membranes. Figure 2 and Figure 3 show the FTIR spectra of the modified membranes from experiments 1 and 2, respectively. The membranes were described as PSU-VIM according to VIM concentration.

Figure 2 also included spectra of PSU film and spectra of VIM (50%). As a result, the band at 3104 cm^−1^ from VIM was attributed to the -CH group from the ring. Bands from -CH_2_ and -CH_3_ groups, at 2950 cm^−1^, were also observed [43]. In addition, distinctive bands, at 1645 cm^−1^, indicated the stretch vibration of the vinyl aromatic groups (C=C); whereas, bands at 1510 and 1492 cm^−1^ described the aromatic C=N and C-N stretching bonds. Furthermore, bands at 1224 cm^−1^ and 728 cm^−1^ were related to the N-C-N and the –C-N bonds [44,45].

The PSU spectrum of Figure 2 exhibited its characteristic bands at 2989 cm^−1^, corresponding to the -CH_3_ group. Bands at 1587–1483 cm^−1^ were attributed to the benzene rings [46]; whereas, bands at 1236 and 1155 cm^−1^ described the asymmetric and symmetric stretching of the SO_2_ vibration. In addition, the band at 1014 cm^−1^ is associated with the C-O-C group [47,48].

On the other side, the spectra of the PSU grafted membranes (PSU-VIM) with 50% *v/v* concentration and gamma irradiation doses of 200–400 kGy, confirmed the presence of the grafted VIM monomer by its characteristic bands at 3061 cm^−1^, as well as, a band from the vinyl group -C=C- at 1645 cm^−1^. Moreover, a band at 744 cm^−1^ was associated with the -C-N- bond from the VIM structure [28].

Spectra of the grafted membranes at different VIM concentrations (PSU-VIM), with an irradiation dose of 300 kGy, are presented in Figure 3 (from Experiment 2).

The exhibited spectra bands, related to the grafted material, confirmed the modification of the PSU surface with the grafting polymerization of VIM. Herein, it was observed that the grafted membranes presented the characteristic bands from VIM, such as, the -CH group from the ring at 3104 cm^−1^, the asymmetric stretching band of the -CH_2_ and -CH_3_ groups at 2950 cm^−1^, the -C-N stretching band at 728 cm^−1^; the stretching vibration of the –C-N group at 1645 cm^−1^ and the stretching of the C=C from the ring at 1650 cm^−1^.

According to the concentration of VIM (50–60% *v/v*), Figure 3 also displayed an augment in the intensity of the VIM bands, indicating the effect of VIM concentration onto the grafted membranes [2,45,49,50].

Nevertheless, grafted membranes with a VIM monomer concentration of 70% did not reveal this particularity; spectra showed a fuzzy band, that was associated with the increment in the viscosity of the grafting VIM, producing a difficult diffusion onto the PSU membrane [51].

Derived from the above analysis, it was demonstrated that gamma irradiation allowed the grafting process of the VIM monomer onto the surface of the PSU film; which makes it a sustainable process, since the use of an initiator was not necessary for the obtainment of this grafted membrane. Besides, FTIR analysis showed that irradiation doses of 300 kGy increased the grafting yield of VIM onto the PSU membranes, when the solution concentration of the grafting agent was found to be between 50% and 60% (*v/v*).

### 3.3. Surface Morphology of the Grafted PSU Membranes

Figure 4 shows SEM images of the grafted PSU membranes from experiment 1. (a) PSU film, and grafted PSU-VIM membranes with VlM monomer concentration of 50% *v/v*, using gamma irradiation doses of (b) 200, (c) 300 and (d) 400 kGy, and SEM images from Experiment 2 of the grafted PSU-VIM membranes with 300 kGy of irradiation dose and different VIM monomer concentrations at (e) 30%, (f) 50%, (g) 60%, and (h) 70% (*v/v*).

As was expected, PSU film showed a homogeneous surface compared to the grafted membranes. A similar image of a PSU film can be observed in the work reported by Li et al. [52]; however, the membrane surface images of the grafted PSU-VIM membranes displayed different levels of modification because of the employed irradiation doses.

The obtained PSU membranes with 200 kGy showed small alterations distributed along the surface, which was associated with the grafting of VIM monomer, due to the VIM polymerization.

Consequently, the grafted membranes with an irradiation dose of 300 kGy presented heterogeneous surfaces with clump-like overlays, which were more notorious than in the experiment 1 and these were most probably associated with an increment of the VIM concentration and irradiation doses. In turn, the grafted membranes with 400 kGy showed a different morphology surface. The difference was linked to the accumulation of grafted material, due to an increment of the viscosity of the VIM monomer, which was caused by the absorbed dose. The heterogeneous surface formation in PSU grafted membranes was attributed to the VIM homopolymerization, due to the limited mobility of the unreacted monomer molecules of VIM towards polymer radicals, and the deficiency of active groups in the PSU, producing an unfavorable grafting [31,45].

Regarding Experiment 2, SEM images showed that the grafted membrane with VIM monomer at 30% (*v/v*) presented some smaller clumps of graft material larger than 1 µm in size, well distributed along the surface, demonstrating the graft of polymerized VIM; whereas, images of grafted PSU membranes with solutions of VIM monomer at 50% and 60% (*v/v*) displayed the formation of a top rough layer, with similar smaller clumps of material. Accordingly, the roughness of the PSU membrane was produced with an irradiation dose of 300 kGy and a solution of VIM monomer with a concentration between 50% and 60% (*v/v*), as can be seen in Figure 4b,c. Nonetheless, the grafted membrane with a solution of VIM monomer, 70% (*v/v*), exhibited a less homogenous thickness rough surface, made up of accumulated layers of the grafted material on the surface of the PSU membrane, which was probably associated with the combination of the high VIM concentration as well as the irradiation dose.

Accordingly, SEM images from experiment 1 confirmed that the condition of 300 kGy, as the irradiation dose, promoted the graft polymerization of VIM 50% (*v/v*), whereas the 400 kGy dose promoted the accumulation of graft material onto the surface of the PSU-VIM membranes. In addition, images from Experiment 2 complemented the assumption that a dose of 300 kGy indorsed an adequate graft on PSU-VIM membranes, using VIM solution concentrations between 50–60% (*v/v*). Nevertheless, an increment in the heterogeneous surface was observed in grafted PSU-VIM 60% (*v/v*).

Subsequently, PSU-VIM membranes 70% (*v/v*) showed a mainly heterogeneous surface, due to the formation of aggregates of VIM and accumulation on the membrane surface. A high concentration of VIM and the homopolymerization of this grafting agent elucidated this result [53], as has already been explained in Experiment 1.

Moreover, the heterogeneous surface of the grafted membranes could be related to surface roughness. Although AFM was not include in this study, it could be expected that the external irregularity of membranes 60–70% (*v/v*) will probably display a high average roughness in comparison with PSU film, which could affect the membrane operation, since a heterogeneous surface increases the fouling tendency.

The surface roughness, due to the homopolymerization phenomenon of graft material, was also observed in other reports. Meléndez-Ortiz et al. [28] showed a strong VIM homopolymerization process onto silicone rubber for doses above 50 kGy when pure VIM (100%) was used; on the other side, Nguyen et al. [54] and Lopez-Saucedo et al. [55] detected this phenomenon in the grafting process of VIM onto poly(tetrafluoroethylene) films, PTFE, observing that high radiation doses and high concentrations of the grafting material produce a heterogeneous surface with the subsequent formation of aggregates, due to the homopolymerization process of VIM.

### 3.4. Hydrophilicity Properties of the Grafted PSU Membranes

The hydrophilicity properties of grafted membranes PSU-VIM were defined in Table 1 by water contact angle, and in Figure 5 by the swelling degree percentage.

Table 1 shows images and values of contact angles for the grafted PSU-VIM membranes. The information is presented according to experiments 1 and 2 (variating irradiation doses and concentration of the grafting agent, respectively).

Data from experiment 1 indicate that the grafting process of VIM 50% (*v/v*), using different gamma irradiation doses (200–300 kGy), onto PSU membranes affected the contact angle, reducing this value by up to 35% according to the amount of the grafted VIM onto the PSU film. However, the grafted membrane with an irradiation dose of 400 kGy showed a different tendency. This difference was associated with the accumulation of the grafted material onto the membrane surface, as was previously mentioned, causing a similar contact angle than the grafted membrane with 300 kGy. Experiment 2 also exposed the reduced contact angle onto the grafted membranes with different VIM concentrations and an irradiation dose of 300 kGy. The contact angle values indicated that a high concentration of VIM (50–60% *v/v*) promoted the obtention of hydrophilic membranes with small contact angles, reducing their values up to 88% regarding PSU film. However, it was also observable that the contact angle of the grafted membrane with a solution of VIM monomer at 70% (*v/v*) was not detected (ND), indicating a value close to zero. Similarly, this result was linked to the high accumulation of VIM monomer onto the membrane surface.

In general, data from both experiments indicated a tendency of water contact angle reduction as a consequence of the increment in the grafting yield. Therefore, the hydrophilicity of the PSU membrane surface was successfully improved, attributing this result to the contact angle decrease, due to the introduction of the hydrophilic functionality of the VIM ring and to the adequate doses of irradiation, in order to achieve the change in this property [24,41,45].

Complementing the analysis of hydrophilicity, Figure 5 shows the swelling degree behavior of the grafted membranes (from Experiment 2) as a function of pH in the aqueous solution. The swelling degree showed a similar behavior, which is in concordance with the contact angle results. The grafted PSU membranes (50–70% *v/v*) showed higher values of swelling degree, which means a higher water permeation compared with the PSU pristine film. The swelling degree was also associated with a greater hydrophilicity conferred by the VIM grafting agent, due to its water-soluble nature and the reduced hydrogen bond into PSU membrane [41]. Herein, it was notorious that the grafted membrane, with a 60–70% (*v/v*) VIM monomer concentration, presented the highest value of the swelling degree. However, the observed reduction of the mechanical resistance for these grafted membranes exposed that the high swelling degree could be attributed to the formed channels onto the membrane surface, due to the accumulation of the grafted material and the VIM property.

In the case of the swelling behavior as a function of the pH of the solution, the grafted PSU-VIM membranes exposed a swelling phase from a pH value of 2 up to 10.5, displaying high percentages according to the concentration of VIM in the membrane. The swelling behavior of these curves showed also a decrease caused by the pH change from neutral to basic. This shape contraction displayed a critical pH value of 7.7 (inflexion point in the curve), evidencing that the grafted PSU-VIM membranes presented a pH-responsiveness, and thus, an attained water permeation due to the presence of protonated imidazole rings [56].

From these results, the grafted PSU-VIM membrane with an irradiation dose of 300 kGy and VIM 50% *v/v* were considered as promising hydrophilic membranes, with a critical pH value of 7.7.

Moreover, the reduction of the contact angle and the high swelling degree from the grafted PSU membrane indicated an increment in their hydrophilicity properties, exhibiting a smallest fouling tendency. The membrane hydrophilicity was attributed to the hydrophilic nature of the VIM onto PSU membranes. Specifically, the principal change of hydrophilicity was observed in the PSU grafted membranes (50–60% *v/v*), due to an important reduction in the contact angle values and an increment in the swelling degree, without any defects on the membrane, as was previously discussed.

Theoretically, the membrane hydrophilicity is directly related to antifouling properties, because a hydrophilic membrane corresponds to lower membrane fouling potential than a hydrophobic one. Nevertheless, the hydrophilicity is not always verified during the membrane operation, because experimentally there are other factors that affect its hydrophilicity, such as operational pressure and feed characteristics [57]. In consequence different parameters about the membrane hydrophilicity are reported; however, contact angles and the swelling degree are often used as parameters to provide information on fouling behavior.

In this case, the obtained contact angles by grafting VIM onto PSU membranes were not similar to the report of López-Saucedo et al. [55], because their membrane experiments were different. They studied the binary grafting process of VIM and methyl methacrylate (MMA) (with concentrations of 15% and 5%, respectively) onto PTFE films, showing a higher contact angle value (91.9° ± 4.0); however, authors indicated a low hydrophilicity modification, which was attributed to the grafting agent concentration and to the membrane material. Nevertheless, according to the report from Pérez-Calixto et al. [58], the contact angle after the grafting process was consistent with the values found in our work. Authors used allylamine (AA) for its grafting onto PP films by the gamma irradiation technique, producing lower contact angle values (99.7° to 76°) at 100 kGy.

Furthermore, other grafting agents and additional modification techniques have shown different data regarding membrane hydrophilicity by the contact angle technique and some other parameters. Jang et al. [59] used etherificated PVA with monochloroacetic acid and subsequently the application of PVA–OCH_2_COONa, to enhance the hydrophilicity and to provide fouling resistance in a PVDF membrane. The contact angle was reduced to 37% and the fouling tests with bovine serum albumin showed that the modified membrane presented a higher flux of pure water and a retarded decline in the flux over the filtration period, compared with the PVA-coated PVDF membrane. Liu et al. [60] modified an ultrafiltration membrane of PVDF with graphene oxide and a quaternary ammonium salt by the immersion phase method. The modified membrane exhibited an increment of 30% in its hydrophilicity, with an important reduction of the contact angle value. Xiang et al. [61] synthetized a PVDF/polyamide-6 (PA6) membrane via thermally induced phase separation (TIPS). The introduction of PA6 in the PVDF membrane incremented its hydrophilicity, indicating a reduction of 50% in the contact angle value, improving the rejection to petroleum ether-in-water emulsion and congo red (CR), with an enhancement of the anti-fouling capacity.

Comparable data about the swelling degree and the critical pH value also validated the hydrophilicity range of the grafted PSU membranes with the irradiation method. The results of Pino-Ramos et al. [62] confirmed a similar behavior for the hydrophilic grafted membranes. They calculated the critical pH value of films obtained by induced ionizing radiation grafting of 4-vinylpyridine and N-vinylcaprolactam onto silicone rubber. Changes in the swelling degree of membranes showed that the film which had the maximum grafting yield percentage (42%) presented a remarkable pH swelling behavior and a pronounced collapse, showing pH critical values at 7.5 in a range between 3.0 and 10.5, which was attributed to the grafting agents, the 4-vinylpyridine and the N-vinylcaprolactam onto the grafted material.

The hydrophilicity information of the membrane flux for the grafted PSU membranes is also provided in the next section, as the desalination test, which presents results of flux permeation and salts rejection, confirms changes in the hydrophilicity property of the modified membranes.

### 3.5. Thermal Behavior of the Grafted PSU Membranes

Table 2 and Figure 6 and Figure 7 show the thermal behavior obtained from the TGA (range 25 to 900 °C) and from the DSC analysis (range 25–300 °C). Data correspond to the grafted PSU membranes with solutions of VIM monomer at 50–70% (*v/v*) from Experiment 2, showing the mass decomposition and the heat difference. TGA analysis indicated that PSU film exhibited the highest thermal stability (Ts) and thermal decomposition (Td) in the range between 501–529 °C, maintaining a constant weight of 26%; whereas Ts and Td of the grafted membranes were dependent on the VIM concentration.

In the case of VIM concentration (50–70%, *v/v*), Td of the grafted membranes were 401, 368 and 310 °C, respectively; whereas, Ts were 514, 513 and 510 °C, respectively; obtaining a weight of 0.08%; which was linked to the reduced crystal arrangement in the chemical structure of the PSU [63].

The range of Td for the grafted membranes was analogous to other reports, i.e., Zhang et al. [18]; Meléndez-Ortiz et al. [28]; Abdul Mannan et al. [64]; indicated a Td range between 310 and 480 °C.

In addition, other authors showed different Ts and Td values; however, the discrepancies were associated with the grafting agent and the grafting method. Kutcherlapati et al. [43] proved that Ts value decreased (450–425 °C) and the weight loss augmented, in concordance with the increment of the grafted PNVIM onto the silica nanoparticles, using the RAFT method; Bai et al. [65] also grafted PNVIM onto a PSU backbone via atom transfer radical polymerization (ATRP), obtaining a Td value above 258 °C.

Furthermore, glass transition temperatures (Tg) for PSU film were found at 189.15 °C. This value was in concordance with those reported elsewhere (190–200 °C) [66,67,68]. Sequentially, PSU-VIM membranes displayed a Tg range of 185.40 to 186.58 °C. The Tg reduction from PSU-VIM 50% (*v/v*) was associated with the presence of VIM onto the PSU-film [69], indicating a successful membrane grafting. However, PSU-VIM (60–70% *v/v*) exhibited a different Tg behavior, which was associated with the VIM accumulation onto membranes’ surfaces.

A similar Tg behavior was found by Woo et al. [70]; they observed a decrement of the Tg value for the grafted PSU membranes with polyethylene glycol. In turn, Mushtaq et al. [71] reported that the grafting did not affect the thermal stability of polymeric blend membranes.

On the other side, Figure 7 presents the DSC thermograms for the grafted PSU membranes, with an irradiation dose of 300 kGy and a VIM concentration between 50–70% (*v/v*).

DSC thermograms for the first heating showed an endothermic peak for the PSU-VIM membranes between 80 and 86 °C, which were identified as the loss of some molecules of water which were entrapped within the structure of the membrane; meanwhile, thermograms displayed the characteristic changes in curvature for the thermal behavior, indicating the modification of the Tg for membranes.

### 3.6. Desalination Test for the Grafted PSU Membranes

Table 3 presents data from the rejection permeation of saline solutions (1000 mg/L NaCl used as feed membrane flow at pH 6.6) for a desalination process using the modified PSU-VIM membrane from Experiment 2 (irradiation dose of 300 kGy, and VIM concentration of 50–70% (*v/v*).

Because of the film structure, the used feed flow and total solids (NaCl concentration) were lower (previous trials) than the feed flow for the PSU-VIM 50–70% *v/v*; therefore, the rejection and permeation flows were also in concordance with the membrane samples.

Specifically, the desalination test showed that rejection-permeation of the PSU-VIM (50% *v/v*) was augmented eight times more than that of the PSU pristine film. In addition, the permeation-rejection property was also incremented. Therefore, it was established that the grafting process indorsed the PSU film modification to obtain PSU-VIM membranes 50% (*v/v*), with the highest capacity for salt rejection and permeability, due to their hydrophilicity properties [72,73].

In concordance with the swelling degree data, the pH of feed solution contributed to enhance permeation-rejection results from this membrane, because the range of pH 3.0–6.5 showed the highest values of the swelling degree. Consequently, a drastic reduction in permeation-rejection property above pH of 7.7 is expected, since the swelling showed a critical pH at this value, indicating a depression in the hydrophilicity property.

In turn, grafted membranes PSU-VIM 60–70% (*v/v*) displayed predictable results in desalination tests, confirming two aspects mentioned in the previous information. During the desalination process, the membranes showed a noticeable degradation and breaking, due to pressure operation, thereby affecting the membrane permeability. In consequence, short times of operation and low salt rejection capacity were observed. Both results were attributed to a low mechanical resistance and channels formation, due to the excess of VIM onto PSU and membrane degradation by irradiation exposure in visible areas of grafted membranes.

According to the above data, it was established that grafting conditions of 300 kGy and VIM 50% (*v/v*) were adequate to obtain grafted PSU membranes with high hydrophilicity and functionality for desalination processes.

Consistent results of the modified membranes by surface grafting were found in the manuscript of Deng et al. [74]. Authors used poly(vinylidene fluoride) (PVDF) powder and acrylic acid (AAc) or methacrylic acid (MAAc) to prepare microfiltration membranes with antifouling properties via the pre-irradiation induced graft polymerization technique. The grafted membranes exceeded, by 20%, the water flux performance compared to the pristine membrane. Reis et al. [75] modified the surface charge of PA membranes, demonstrating that conditions of 1–10 kGy and the VIM concentration of 1% *v/v* in methanol/water (1:1) solutions can alter the net surface charge of the pristine membranes from −25 to +45 mV and its isoelectric point from a pH value of 3 to 7, enhancing the water flux over 55%, whereas NaCl rejection was found to drop by only 1% compared to the pristine membranes. The difference in the membrane capacity was attributed to the surface membrane structure and its properties.

## 4. Conclusions

Experiments variating the gamma irradiation doses and VIM concentrations in the preparation of grafted PSU membranes indicated that irradiation doses of 300 kGy and VIM concentrations >50% *v/v* promoted the grafting process, obtaining yields from 14% to 20%; whereas conditions <300 kGy and VIM <50% *v/v* did not promote the VIM grafting. The grafted VIM onto PSU was demonstrated with FTIR analysis and SEM images. In this case, SEM images of the grafted membranes showed different degrees of surface modification, according to the grafting conditions. Nevertheless, it was demonstrated that higher irradiation doses >300 kGy and VIM concentrations >50% caused surface accumulation of VIM and PSU degradation, which affects the mechanical resistance of the membrane surface.

The water contact angle and the swelling degree measurements indicated that the grafting process with the VIM monomer incremented the hydrophilic properties of the grafted membranes. In addition, TGA and DSC analysis of the grafted PSU membranes indicated that the grafting and the irradiation doses did not affect their thermal behavior and resistance. Additionally, desalination tests demonstrated that the grafted membrane, with a solution of VIM monomer at 50% (*v/v*) and 300 kGy of irradiation dose exhibited high flux and salts rejection-water permeation, displaying an increment of 85% of salts rejection compared with the PSU pristine film. In contrast, grafted membranes PSU-VIM (60–70% *v/v*) showed a short time of operation and low capacity of salt rejection permeation, due to accumulation of graft agent and membrane degradation caused by irradiation doses. Finally, according to the obtained results, gamma irradiation could be considered to produce grafted PSU membranes for the desalination process, which may well be competitive with membranes for the OI process.

## Figures and Tables

**Figure 1 polymers-12-01284-f001:**
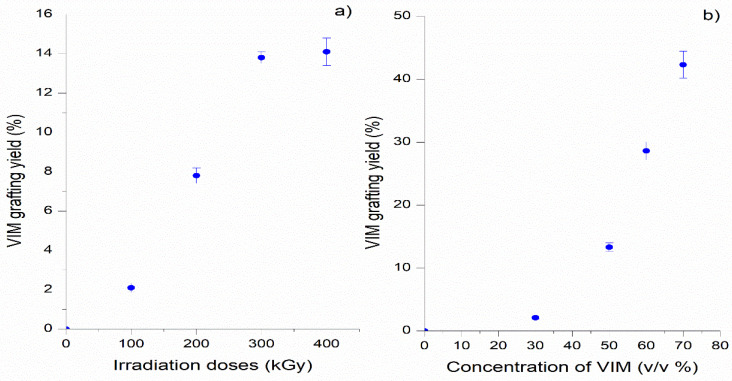
Grafting yield (%) of N-vinylimidazole (VIM) VIM onto polysulfone (PSU) membranes with different gamma irradiation doses: (**a**) experiment 1 (VIM concentration 50% *v/v* and gamma irradiation doses between 100–400 kGy); (**b**) experiment 2 (irradiation dose of 300 kGy and VIM concentration between 30–70% *v/v*). Standard deviation of 3 samples.

**Figure 2 polymers-12-01284-f002:**
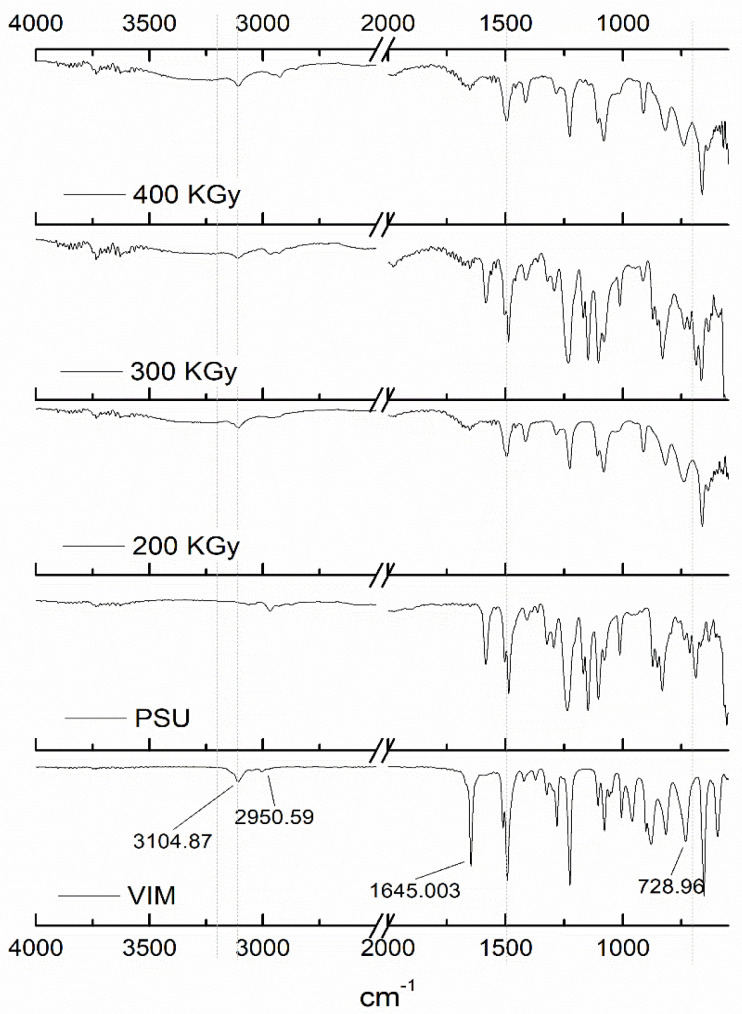
FTIR spectra of grafted PSU membranes with VIM 50% (*v/v*) and gamma radiation at different irradiation absorption doses: 200, 300 and 400 kGy.

**Figure 3 polymers-12-01284-f003:**
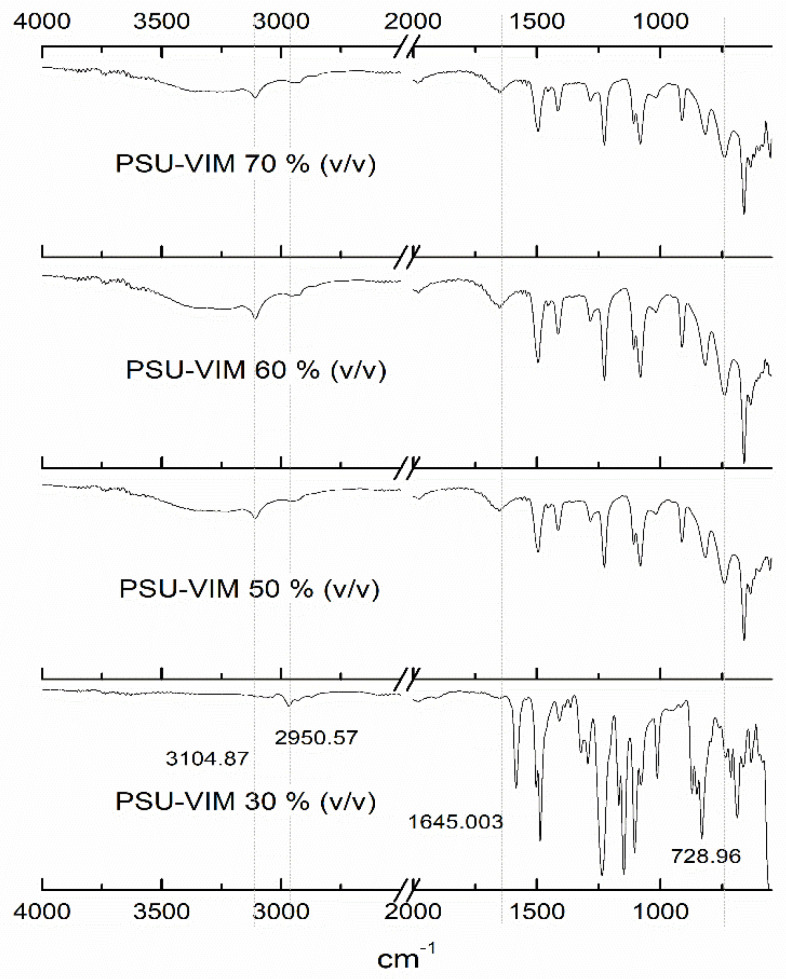
FTIR spectra of the grafted PSU membranes with VIM at different concentrations 30–70% and gamma irradiation absorption dose of 300 kGy.

**Figure 4 polymers-12-01284-f004:**
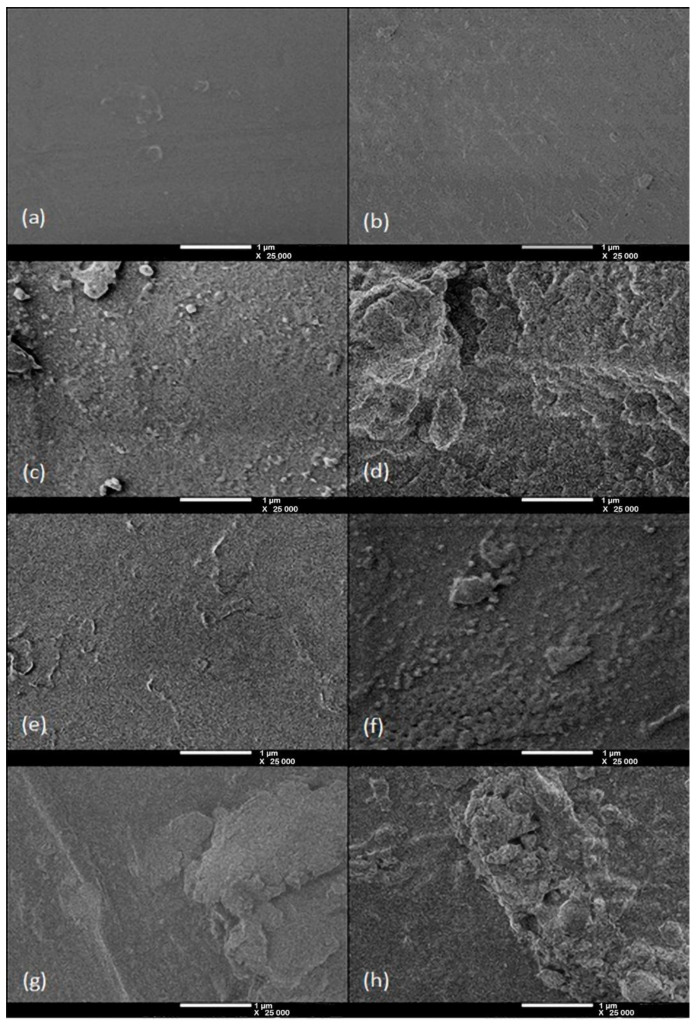
SEM images taken at 25,000× and 0.50 kV from (**a**) PSU film, and PSU grafted membrane, with 50% (*v/v*) of VIM at (**b**) 200, (**c**) 300, and (**d**) 400 kGy. PSU grafted membrane at 300 kGy and (**e**) 30%, (**f**) 50%, (**g**) 60% and (**h**) 70% (*v/v*) of VIM.

**Figure 5 polymers-12-01284-f005:**
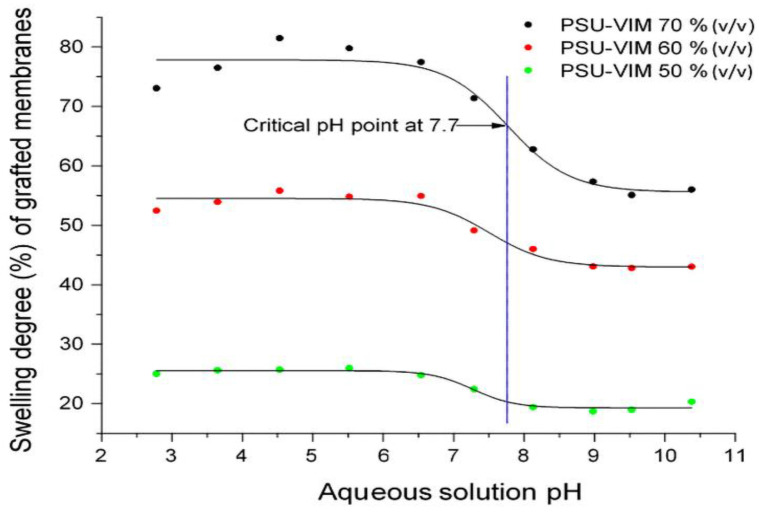
Swelling degree percentage (%) as a function of pH values for the grafted PSU membranes, with an irradiation dose of 300 kGy and VIM monomer concentrations of 50–70% (*v/v*), from Experiment 2.

**Figure 6 polymers-12-01284-f006:**
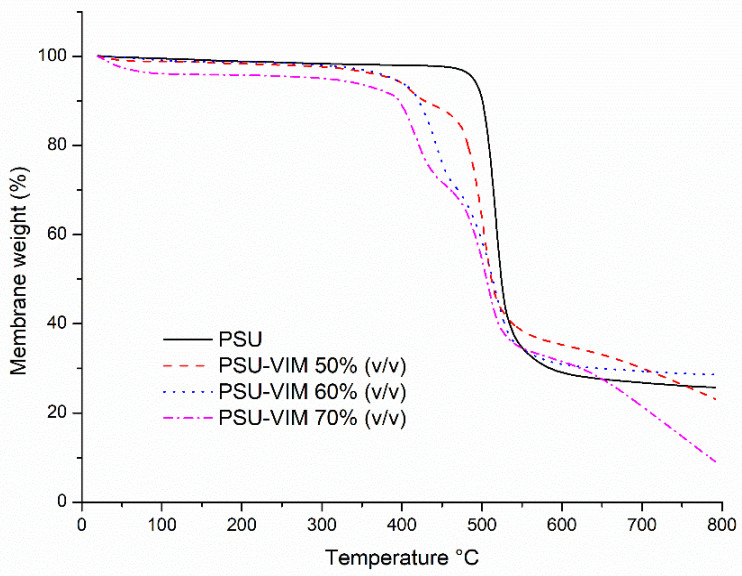
Thermal behavior obtained from TGA analysis for the grafted PSU membranes with an irradiation dose of 300 kGy and VIM concentrations between 50–70% (*v/v*).

**Figure 7 polymers-12-01284-f007:**
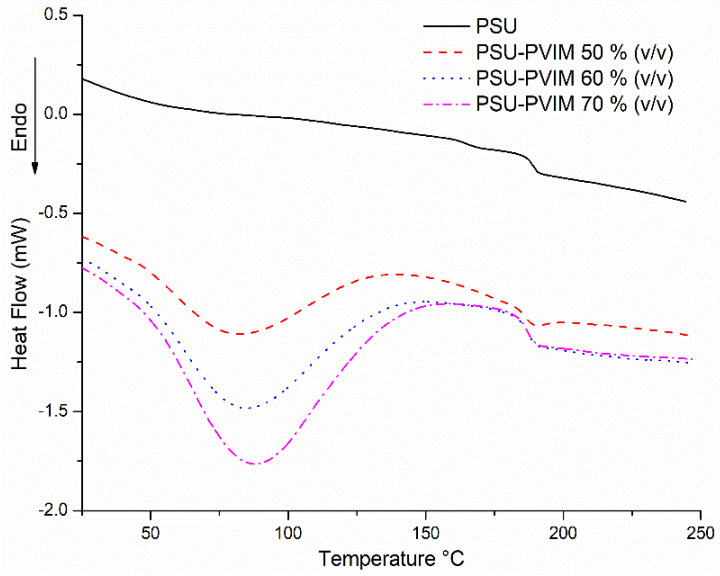
Thermal behavior obtained from DSC analysis, for the first heating scan of the grafted PSU membranes, with an irradiation dose of 300 kGy and VIM concentration between 50–70% (*v/v*).

**Table 1 polymers-12-01284-t001:** Water contact angles of PSU membranes from experiments 1 and 2.

**Experiment 1**	**PSU-VIM 50% (*v/v*) and different gamma irradiation doses (kGy)**
PSU film	200	300	400
Contact angle (^o^)	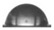	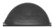	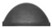	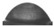
90.7 ± 1	79.2 ± 1.5	66 ± 1.5	64.3 ± 1
**Experiment 2**	**PSU-VIM (30–70% *v/v*) and 300 kGy irradiation dose**
30	50	60	70
Contact angle (^o^)	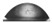	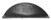	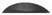	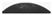
84 ± 0.5	61 ± 0.5	11.8 ± 1	≈ 0

**Table 2 polymers-12-01284-t002:** Data of the thermal behavior of grafted PSU membranes with a solution of VIM monomer at 50%, 60% and 70% *v/v*, with 300 kGy of irradiation dose.

Membrane Sample	Thermogravimetric Analysis (TGA)		Differential Scanning Calorimetry (DSC)	
	Ts (°C)	Td (°C)	Sample Weight (%)	Tg (°C)
PSU film	501	529	26	189.15
PSU-VIM (50% *v/v*)	401	514	13.84	185.40
PSU-VIM (60% *v/v*)	368	513	0.06	186.28
PSU-VIM (70% *v/v*)	310	510	0.08	186.58

**Table 3 polymers-12-01284-t003:** Data of constant flows and total solids from the desalination test of the grafted PSU-VIM (50–70% *v/v*).

Grafting in PSU-Film by VIM	Operation Time (s)	Feed Flow (×10^−4^ mL/cm^2^·s)	Total Solids Feed Flow (mg/L)	Salt Rejection Flow (×10^−4^ mL/cm^2^·s)	Total Solids Rejection Flow (mg/L)	Water Permeation Flow (×10^−4^ mL/cm^2^·s)	Total Solids Permeation Flow (mg/mL)
PSU film	120 ± 15	3.50 ± 1	100 ± 1	1.05 ± 0.3	73.50 ± 5	1.75 ± 0.3	5.87 ± 1
PSU-VIM 50% *v/v*	900 ± 10	24.5 ±3	1000 ± 5	7.36 ± 7	845.00 ± 15	15.18 ± 3	85.34 ± 5
PSU-VIM 60% *v/v*	250 ± 15	29.1 ± 5	1000 ± 5	3.55 ± 5	292.15 ± 21	16.76 ± 2	488.65 ± 5
PSU-VIM 70% *v/v*	168 ± 5	30.2 ± 10	1000 ± 5	2.34 ± 0.9	125.78 ± 25	25.95 ± 10	595.21 ± 20

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
