# Peer review of "Obtainment and Characterization of Hydrophilic Polysulfone Membranes by N-Vinylimidazole Grafting Induced by Gamma Irradiation"

_polymers, 2020, doi:10.3390/polym12061284_

Round 1

Reviewer 1 Report

The work reported in this manuscript (polymers-799572) is interesting and well presented. However, it needs improvements before acceptance. The work requires minor revision. Comments are: ‎

Comment 1: The SEM images (Figure 4) scale bar is not properly visible, so authors needs to draw the scale bars manually to attain more clearly.  

Comment 2: The AFM analysis also needs to be carried out to identify the membranes surface properties.

Comment 3: There are some typographical errors in the manuscript, so authors need to correct it in the revised manuscript.

Comment 4: What is the key impact of hydrophilic properties? Please compare with other similar reported membranes.

Author Response

Response to reviewers

Metepec, Mexico May 13th, 2020

Managing Editor

POLYMERS

Dear Managing Editor

Herein, we are resubmitting our manuscript entitled:

“OBTAINMENT AND CHARACTERIZATION OF HYDROPHILIC POLYSULFONE MEMBRANES BY N-VINYLIMIDAZOLE GRAFTING INDUCED BY GAMMA IRRADIATION”

Manuscript ID: polymers-799572.

Elizabeth Vázquez, *Claudia Muro, Javier Illescas, Guillermina Burillo, Omar Hernández, Ernesto Rivera.

The authors are deeply thankful for all of the observations and suggestions expressed by the reviewers to improve the quality of this work. Our response for each of the comments/queries are enumerated below.

Referees comments to Author

Comment 1: The SEM images (Figure 4) scale bar is not properly visible, so authors need to draw the scale bars manually to attain more clearly.

Response: We thank the reviewer for his observation. The Figure 4 was improved and scales for SEM images were placed for each Figure.

Comment 2: The AFM analysis also needs to be carried out to identify the membranes surface properties.

Response: We deeply appreciate to the reviewer for this observation. We have planned to perform some AFM experiments to check out the membrane surface properties. Unfortunately, since we are on lockdown because of the pandemic we are unable to perform any significant experiments since all institutions in Mexico are closed. This could delay the publication of our communication, and we do think that with the information obtained and presented from the different characterization techniques within this manuscript, it is enough to prove the membrane surface modifications and their properties.

Comment 3: There are some typographical errors in the manuscript, so authors need to correct it in the revised manuscript.

Response: We also thank the reviewer for this comment. The manuscript was reviewed and corrected by all of the authors, maintaining the original text idea and concordance between experiments and results.

Comment 4: What is the key impact of hydrophilic properties? Please compare with other similar reported membranes.

Response: We thank again the reviewer for this clarification. Authors introduced a paragraph in the section 3.4 of the manuscript, “Hydrophilicity properties of the grafted PSU” in order to explain the impact of hydrophilicity properties in the membranes, and also, to compare results with other authors (more references were introduced). The paragraph was remarked with yellow colour in the manuscript.

Finally, we are very grateful to reviewers since their observations, without any further questions, helped to improve the quality of this manuscript.

We look forward to satisfying the standard conditions established by POLYMERS journal.

Best regards,

C. Muro

Reviewer 2 Report

The work deals with an interesting problem of obtaining of grafted membranes with high hydrophilic capacity via using of polysulfone film and N-vinylimidazole (VIM) by combining different VIM concentrations and gamma irradiation doses. The effect is quite thoroughly characterized with a set of appropriate experimental techniques. Different approaches applied by the authors yield mainly the same result with a certain difference relating to specific irradiation doses and VIM concentration. The work definitely has a technological impact.
The strength of the work: Detail characterization of the objects by a set of complementary spectroscopic and structure/morphology sensitive techniques.
The weakness: The work would gain if more physico-chemical insights into the process are provided, for instance closer interrelation between information obtained by different techniques.​
The work is clearly and accurately presented, lacks of errors, figures are informative and of good quality, reference list is appropriate and up-to-dated.
The work is suitable for publication in Polymers, and can be published in its present form.

Author Response

Response to reviewers

Metepec, Mexico May 13th, 2020

Managing Editor

POLYMERS

Dear Managing Editor

Herein, we are resubmitting our manuscript entitled:

“OBTAINMENT AND CHARACTERIZATION OF HYDROPHILIC POLYSULFONE MEMBRANES BY N-VINYLIMIDAZOLE GRAFTING INDUCED BY GAMMA IRRADIATION”

Manuscript ID: polymers-799572.

Elizabeth Vázquez, *Claudia Muro, Javier Illescas, Guillermina Burillo, Omar Hernández, Ernesto Rivera.

The authors are deeply thankful for all the observations and suggestions expressed by the reviewers to improve the quality of this work.

We look forward to satisfying the standard conditions established by POLYMERS journal.

Best regards,

C. Muro
